# Deep Learning-Based Upper Limb Functional Assessment Using a Single Kinect v2 Sensor

**DOI:** 10.3390/s20071903

**Published:** 2020-03-30

**Authors:** Ye Ma, Dongwei Liu, Laisi Cai

**Affiliations:** 1Research Academy of Grand Health, Faculty of Sports Science, Ningbo University, Ningbo 315000, China; 2School of Information, Zhejiang University of Finance and Economics, Hangzhou 310018, China; dongwei.liu@zufe.edu.cn; 3Faculty of Sports Science, Ningbo University, Ningbo 315000, China; cailaisi1995@163.com

**Keywords:** upper limb functional assessment, Kinect, deep learning, recurrent neural network, kinematics

## Abstract

We develop a deep learning refined kinematic model for accurately assessing upper limb joint angles using a single Kinect v2 sensor. We train a long short-term memory recurrent neural network using a supervised machine learning architecture to compensate for the systematic error of the Kinect kinematic model, taking a marker-based three-dimensional motion capture system (3DMC) as the golden standard. A series of upper limb functional task experiments were conducted, namely hand to the contralateral shoulder, hand to mouth or drinking, combing hair, and hand to back pocket. Our deep learning-based model significantly improves the performance of a single Kinect v2 sensor for all investigated upper limb joint angles across all functional tasks. Using a single Kinect v2 sensor, our deep learning-based model could measure shoulder and elbow flexion/extension waveforms with mean CMCs >0.93 for all tasks, shoulder adduction/abduction, and internal/external rotation waveforms with mean CMCs >0.8 for most of the tasks. The mean deviations of angles at the point of target achieved and range of motion are under 5° for all investigated joint angles during all functional tasks. Compared with the 3DMC, our presented system is easier to operate and needs less laboratory space.

## 1. Introduction

Three dimensional (3D) kinematic analysis of upper limb functional movement has been widely conducted in many areas. Upper limb kinematic analysis has been employed in both theoretical studies such as the underlying theory of neuromusculoskeletal system [1,2,3] and practical concerns in the clinical assessment of motion functions, rehabilitation training [4], ergonomics studies [5,6], and so forth. *Marker-based 3D motion capture systems* (3DMC) [7] have been widely employed in quantitative measurements of upper limb functional tasks. In such a system, 3D motion data is obtained based on passive or active markers attached to the anatomical landmarks of participants. These marker-based systems have been confirmed to be valid and reliable in assessing upper limb kinematics [3,8]. However, these systems are not practical for applications in small clinics or home-based assessment, given the expensive hardware cost, time-consuming experiment conduction as well as the strict requirements for lab space and trained technician.

Markerless motion capture system could be a possible alternative for upper limb functional assessment [9], especially after the introduction of a commercially available, low-cost, and portable device named Kinect (Microsoft, Redmond, WA, USA). The second iteration of the Kinect (denoted as Kinect v2), is capable of tracking real-time 3D motions with its depth image sensor [10] and its human pose estimation algorithm [11]. The Kinect SDK v2.0 features skeletal tracking with 3D locations of 25 joints for each skeleton [12]. Kinect v2 has been employed in gait analysis [13,14,15], balance and postural assessment [16,17], foot position tracking [18], gait rehabilitation training [19,20], upper limb functional assessment or rehabilitation training [4,21,22,23,24,25].

Several studies have assessed the agreement between Kinect sensor and 3DMC. Kinect sensor demonstrated good reliability in assessing temporal-spatial parameters such as timing, velocity, or movement distance of functional tasks for both healthy subjects and people with physical disorders [4,13,21,22,26,27]. Kinect also has considerable good reliability in kinematics assessment such as upper limb joint angle trajectories [22,28] and the respective range of motions [28], trunk flexion angles during the standing and dynamic balance test [16], trunk, hip, and knee kinematics during squatting and jumping tasks [29] or foot postural index assessment [30].

Kinect sensor has been employed in various rehabilitation scenarios for people with motor disabilities [23,25]. A Kinect-based rehabilitation system improved exercise performance for adults with motor impairment during the intervention phase [23]. An RGBD-based interactive system using the Kinect sensor provided a gamified interface designed to replace physiotherapists in the supervision of upper limb exercises. The interactive system is able to provide real-time feedback and to create interactive, simple to use and fun environments to patients. The system improves the engagement of the participants and the effectiveness of the exercises [25].

Nevertheless, Kinect is limited by its inherent inaccuracy. Although the system has good accuracy in measuring temporal-spatial parameters such as the timing of movements [4], velocity, or distance measurement during movement [26], gross spatial characteristics of clinically relevant movements such as vertical oscillation during treadmill running [31], it lacks enough accuracy in small movements, such as hand clasping [4].

The Kinect system captures RGBD data with a time-of-flight sensor [10]. Such RGBD data is a 2D image with depth information for each pixel, which is not a complete 3D model. Intuitively speaking, the RGBD data can be seen as a “relieve”, or a “2.5D model”. Skeletons calculated from such RGBD data suffer from certain systematic errors due to the inaccurate depth measurement. The Kinect system is highly task-dependent and plane-dependent in terms of kinematic measurement accuracy [28]. Due to the “relieve” feature, the Kinect system measures more accurate joint angles on the sagittal plane and coronal plane than on the transverse plane [28,32,33]. 

The performance of the Kinect system is highly influenced by the occlusion of body parts. The Kinect camera cannot directly assess necessary anatomical joint centers if another body part is in between with the segment and the camera. The Kinect system is mostly placed in front of the human subjects. In this scenario, assessing upper limb functional tasks is more challenging than conducting gait analysis for the Kinect system due to the high probability of upper limb occlusion when performing upper body activities such as drinking or combing hair [34].

In comparison with the joint angles measured with 3DMC systems, those angles calculated via the Kinect system generally has unacceptable accuracy in clinical assessment. The performance of the Kinect system is normally demonstrated with the root-mean-squared error (RMSE) between the investigated parameter via the Kinect system and the 3DMC system. The RMSE of the Kinect based system is around 10° in measuring shoulder flexion/extension, 10° to 15° in measuring adduction/abduction, or approximately 15° to 30° in measuring shoulder axial rotation during computer-using task [33]. For people with Parkinson’s disease during multiple reaching tasks and tasks from the Unified Parkinson’s Disease Rating Scale, the mean bias for shoulder flexion/extension, shoulder adduction/abduction and elbow flexion/extension angles between the Kinect and 3DMC system are 10.44°, 8.68° and 16.93° for healthy subjects and 17.07°, 10.26°, and 21.66° for patients [4]. A study of five gross upper body exercises from the GRASP manual for stroke rehabilitation [35] revealed that the Kinect v2 is mostly inadequate for correctly assessing shoulder joint kinematics during stroke rehabilitation exercises. The movements of the shoulder joints are used as indicators for incorrect limb movements. Unacceptable jitter and tracking occurred when the depth data surrounding the joint is partial or completely occluded.

Various researches have been conducted to improve the accuracy of Kinect v2 in kinematics measurement. One type of solution is model fitting algorithms. Xu et al. [33] employed a linear regression algorithm between each Kinect-based shoulder joint angle and its 3DMC counterpart. Given the nonlinear relationship between the upper limb joint angle trajectories calculated via Kinect and the 3DMC system, linear regression algorithms have limited ability to improve the kinematic measurement accuracy. Only shoulder adduction/abduction angles are significantly improved after employed the linear regression algorithms and the RMSEs between the Kinect sensor and the 3DMC system is around 8.1° and 10.1° for the right and left shoulders. Kim et al. [36] proposed a post-processing method, which is a combination of two deep recurrent neural networks (RNN) and a classic Kalman filter, to correct unnatural tracking movements. This post-processing method only improves the naturalness of the captured joint trajectories. The accuracy is insufficient for clinical assessment.

Another type of solution is applying marker-tracking technology with the Kinect system. Timmi et al. [37] developed a novel tracking method using Kinect v2 by employing custom-made colored markers and computer vision techniques. The markers with diameters of 38 mm were painted using matte acrylic paints. Magenta, green, and blue paints were chosen for hip, knee, and ankle joint markers, respectively. The centers of the three markers should be placed on a straight line. They evaluated the method in comparison with lower limb kinematics over a range of gait speeds and found generally good results. However, the actual use case for this kind of system appears limited due to two factors: (1) The marker-tracking Kinect system could not solve the occlusion issue when performing upper limb functional task; (2) The introduction of markers into Kinect system bring the reliability issue from incorrect marker placement and complicated experimental calibration procedures. Thus, the method is unlikely to provide significant benefits over the skeleton tracker algorithm [34].

Using multi-Kinect and fusion systems might be another solution to improve the assessment accuracy of the Kinect system as it can reduce body occlusion and extend the field of view. However, these systems show apparent limitations such as: (1) It is difficult to set up and calibrate multiple depth cameras; (2) One Kinect is likely to be impacted by another Kinect sensor. For this matter, the evidence of improving accuracy is not strong [38].

Given the pros and cons of the existing algorithms, as shown above, we develop a novel *deep learning refined kinematic model* using a single Kinect v2 sensor for accurately assessing 3D upper limb joint angles. We form a kinematic model to calculate upper limb joint angles from Kinect. For a specific task, we construct a deep neural network to compensate for the systematic error on those joint angles. Such a neural network is trained using joint angles via the Kinect sensor as the input and those 3DMC counterparts as the target. For the 3DMC, a *UWA kinematic model* [39] is used to calculate 3D upper limb kinematics based on the 3D positions of reflective markers attached on the subjects. A deep neural network is a favorable tool for non-linear fitting, especially when the shape of the underlying model is unknown [40]. The *recurrent neural network* (RNN) architecture is designed specifically for time series data, which conforms to our joint angel data very well. Long short-term memory (LSTM) is the cutting-edge technology of RNN [41]. We employ a three-layer LSTM network in our method. See Figure 1 for a brief pipeline of our method.

A series of upper limb functional task experiments were conducted to evaluate the effectiveness of our developed deep learning-based model. The tasks represent a variety of active daily functional activities [42]. The hand to contra lateral shoulder task represents activities such as washing axilla or zip up a jacket. The hand to mouth task represents eating or reaching the face. The combing hair task represents washing/combing hair or reaching the back of the head. The hand to back pocket task represents reaching the back and perineal care.

3D positions of the reflective markers according to the UWA marker set are recorded using a 3DMC system. The joint centers extracted from the Kinect skeleton are recorded and a single Kinect v2 sensor. We use a *leave-one subject-out cross-validation* protocol to evaluate the performance of our deep learning refined kinematic model. The *coefficient of multiple correlation* (CMC) and *root-mean-squared error* (RMSE) are used to evaluate the performance of the deep learning refined kinematic model in assessing upper limb angular waveforms in comparison with the kinematic model for Kinect sensor. *Range of motion* (ROM) and *angles at the point of target achieved* (PTA) are extracted to represent key kinematic parameters. ROM and PTA via both our deep learning refined kinematic model and the kinematic model for Kinect sensor are statistically compared with those via the 3DMC system. The absolute error and *Bland-Altman plot* are analyzed for the ROM and PTA via the deep learning refined kinematic model as well as the kinematic model for the Kinect sensor in comparison with those via the 3DMC system.

Our deep learning refined kinematic model significantly improves the performance of upper limb kinematic assessment using a single Kinect v2 sensor for all investigated upper limb joint angles across all functional tasks. At the same time, such an assessment system is also easy to calibrate and operate. The requirements for laboratory space and specialties are easy to be fulfilled for a single Kinect-based system. The system has great potential to be an alternative of the 3DMC system and be widely used in clinics or other organizations, which lacks money, specialties, or lab space.

## 2. Methods

We denote the kinematic model for Kinect by Φ and the UWA kinematic model for a 3DMC system by Γ. The deep learning refined kinematic model for Kinect v2 is denoted by $\hat{\Phi }$, which is a combination of the model Φ and the trained neural network *N*. The upper limb kinematics calculated by model Φ and Γ are defined as $k_{\Phi }$ and $k_{\Gamma }$, respectively. We train a long short-term memory (LSTM) recurrent neural network (RNN) *N* using a supervised machine learning architecture to compensate for the systematic error of Φ. During the training session, $k_{\Phi }$ and $k_{\Gamma }$ are taken as the input data and the target data, respectively. In the application stage, $k_{\Phi }$ is given as the input of *N*, and output is our refined upper limb kinematics (defined as $k_{\hat{\Phi }}$). See Figure 1 for a simple demonstrate.

The UWA kinematic modeling for the 3DMC system and the upper limb kinematic modeling for the Kinect v2 system follow the procedures demonstrated in Figure 2. A standard 3D kinematic modeling procedure [43] includes four steps, namely setting up a global coordination system, setting up local segments coordination systems, calculation of transformation matrix for segment investigated and calculation of upper limb kinematics. The 3DMC system and the Kinect v2 sensor capture 3D marker trajectories and record 3D joint trajectories of a participant concurrently when performing upper limb functional tasks.

### 2.1. Upper Limb Kinematic Modeling for Kinect v2

The 3D coordinates of the anatomical landmarks identified from the skeletal model of the Kinect v2 system (see Figure 3) during functional tasks are recorded concurrently with the 3DMC system. Local segment coordinates, including Thorax λ and Upper Arm η, are established. Each of the segment is based on the global coordinate.

The origin of the thorax segment is defined by SpineShoulder (SS). The y-axis of the thorax segment is defined by the unit vector going from SpineMid (SM) to SS (Equation (1)). The z-axis of the thorax segment is defined by the unit vector perpendicular to y-axes and the vector from ShoulderLeft (SL) to ShoulderRight (SR) (Equation (2)). The x-axis of the thorax segment is defined by z and y-axes to create a right-hand coordinate system (Equation (3)). The coordinate system of the thorax segment $C_{\Phi ,\lambda }$ is then constructed by x, y and z-axis (Equation (4)): (1)$$y_{\Phi ,\lambda }=\frac{\mathrm{SS}-\mathrm{SM}}{\left\|\mathrm{SS}-\mathrm{SM}\right\|}$$
(2)$$z_{\Phi ,\lambda }=\frac{y_{\Phi ,\lambda }\times \left(\mathrm{SR}-\mathrm{SL}\right)}{\|y_{\Phi ,\lambda }\times \left(\mathrm{SR}-\mathrm{SL})\right\|}$$
(3)$$x_{\Phi ,\lambda }=\frac{y_{\Phi ,\lambda }\times Z_{\Phi ,\lambda }}{\left\|y_{\Phi ,\lambda }\times Z_{\Phi ,\lambda }\right\|}$$
(4)$$C_{\left(\Phi ,\lambda \right)}=\left[x_{\Phi ,\lambda },y_{\Phi ,\lambda },z_{\Phi ,\lambda }\right]$$

The origin of the right upper arm segment is the right elbow joint center ElbowRight (ER). The y-axis of the right upper arm segment is defined by the unit vector going from the elbow joint center to shoulder joint center, ShoulderRight (SR), see Equation (5). The z-axis of the right upper arm segment is defined by the unit vector perpendicular to the plane formed by y-axis of the upper arm and the long axis vector of the forearm, pointing laterally (Equation (6)). The x-axis of the right upper arm segment $R_{\Phi ,\eta }$ is defined by the unit vector perpendicular to the z and y-axes, pointing anteriorly (Equation (7)). The coordinate system of the upper arm segment $C_{\Phi ,\eta }$ is then constructed by x, y and z-axis of the segment (Equation (8)): (5)$$y_{\Phi ,\eta }=\frac{\mathrm{SR}-\mathrm{ER}}{\left\|\mathrm{SR}-\mathrm{ER}\right\|}$$
(6)$$z_{\Phi ,\eta }=\frac{y_{\Phi ,\eta }\times \left(\mathrm{ER}-\mathrm{WR}\right)}{\left\|y_{\Phi ,\eta }\times \left(\mathrm{ER}-\mathrm{WR}\right)\right\|}$$
(7)$$x_{\Phi ,\eta }=\frac{y_{\Phi ,\eta }\times z_{\Phi ,\eta }}{\left\|y_{\Phi ,\eta }\times z_{\Phi ,\eta }\right\|}$$
(8)$$C_{\Phi ,\eta }=\left[x_{\Phi ,\eta },y_{\Phi ,\eta },z_{\Phi ,\eta }\right]$$

Then our customized upper limb kinematics model for the Kinect v2 system calculates the three Euler angles ($\alpha_{\mathrm{FE}},\alpha_{\mathrm{AA}},\alpha_{\mathrm{IE}}$) for shoulder rotations, which following the flexion (+)/extension (−), adduction (+)/abduction (−) and internal (+)/external (−) rotation order. The rotation matrix $R_{\Phi }\left(\lambda ,\eta \right)$ is obtained via the parent coordinate system $C_{\Phi ,\lambda }$ (Equation (4)) and the child coordination system $C_{\Phi ,\eta }$ (Equation (8)). Shoulder flexion/extension $\alpha_{\mathrm{FE}}$, adduction/abduction $\alpha_{\mathrm{AA}}$, internal/external rotation $\alpha_{\mathrm{IE}}$ angles are calculated by solving the multivariable equations in Equation (9). (9)$$\begin{array}{c} R_{\Phi }\left(\lambda ,\eta \right)=C_{\Phi ,\lambda }{}^{-1}\times C_{\Phi ,\eta }\\ =[\begin{array}{ccc} -\sin\left(\alpha_{\mathrm{FE}}\right)\sin\left(\alpha_{\mathrm{AA}}\right)\sin\left(\alpha_{\mathrm{IE}}\right)+\cos\left(\alpha_{\mathrm{FE}}\right)\cos\left(\alpha_{\mathrm{IE}}\right) & -\sin\left(\alpha_{\mathrm{FE}}\right)\cos\left(\alpha_{\mathrm{IE}}\right) & \sin\left(\alpha_{\mathrm{FE}}\right)\sin\left(\alpha_{\mathrm{AA}}\right)\cos\left(\alpha_{\mathrm{IE}}\right)+\cos\left(\alpha_{\mathrm{FE}}\right)\sin\left(\alpha_{\mathrm{IE}}\right)\\ \cos\left(\alpha_{\mathrm{FE}}\right)\sin\left(\alpha_{\mathrm{AA}}\right)\sin\left(\alpha_{\mathrm{IE}}\right)+\sin\left(\alpha_{\mathrm{FE}}\right)\cos\left(\alpha_{\mathrm{IE}}\right) & \cos\left(\alpha_{\mathrm{FE}}\right)\cos\left(\alpha_{\mathrm{AA}}\right) & -\cos\left(\alpha_{\mathrm{FE}}\right)\sin\left(\alpha_{\mathrm{AA}}\right)\cos\left(\alpha_{\mathrm{IE}}\right)\\ -\cos\left(\alpha_{\mathrm{AA}}\right)\sin\left(\alpha_{\mathrm{IE}}\right) & \sin\left(\alpha_{\mathrm{AA}}\right) & \cos\left(\alpha_{\mathrm{AA}}\right)\cos\left(\alpha_{\mathrm{IE}}\right) \end{array}] \end{array}$$

The elbow flexion/extension angle $\alpha_{\mathrm{EFE}}$ is calculated by the position data from ShoulderRight (SS), ElbowRight (ER), and WristRight (WR) using the trigonometric function (Equations (10) and (11)). In equation (10), $V_{\Phi ,\mathrm{WE}}$ is the unit vector going from the elbow joint center to the wrist joint center. The upper limb kinematics via the Kinect based system $k_{\Phi }$ is formed by the shoulder and elbow joint angles (Equation (11)). The kinematics model for Kinect v2 was developed using MATLAB 2019a: (10)$$V_{\Phi ,\mathrm{WE}}=\frac{\mathrm{WR}-\mathrm{ER}}{\left\|\mathrm{WR}-\mathrm{ER}\right\|}$$
(11)$$\alpha_{\mathrm{EFE}}=\frac{\mathrm{acos}\left(y_{\Phi ,\eta }\cdot V_{\Phi ,\mathrm{WE}}\right)\times 180}{\pi }$$
(12)$$k_{\Phi }=\left[\alpha_{\mathrm{FE}},\alpha_{\mathrm{AA}},\alpha_{\mathrm{IE}},\alpha_{\mathrm{EFE}}\right]$$

The angular waveforms between the Kinect v2 sensor and the Vicon system are synchronized during post processing. The joint angles from both systems are firstly resampled to 300 Hz using the Matlab function “interp” and then synchronized using a cross-correlation based shift synchronization technique.

### 2.2. UWA Kinematic Modeling via 3D Motion Capture System

The UWA kinematic model Γ for the reference 3DMC system (in this paper we use Vicon, Oxford Metrics Group, Oxford, UK) is based on the 3D trajectories of the reflective markers to the anatomical position of each subject according to the UWA upper limb marker set [44]. The UWA marker set includes the seventh cervical vertebra (C7), 10th thoracic vertebra (T10), sternoclavicular notch (CLAV), xyphoid process of the sternum (STRN), posterior shoulder (PSH), anterior shoulder (ASH), elbow medial epicondyle (EM), elbow lateral epicondyle (EL), most caudal-lateral point on the radial styloid (RS), caudal-medial point on the ulnar styloid (US), a triad of markers affixed to upper arm (PUA), a triad of markers affixed to forearm (DUA) and the metacarpal (CAR) (see Figure 4 for the detailed marker setting). The PUA and DUA are positioned in areas that are not largely influenced by the soft tissue artifact, according to Campbell et al. [45,46]. Medial and lateral elbow epicondyle markers are removed for the dynamic functional tasks.

A biomechanical model is employed based on the UWA upper limb marker set [39,44]. The coordinates of each marker at each sample point in the global coordinate system are recorded and represented by a three-dimensional vector (x, y, z). Four rigid body segments, namely Thorax, Torso, Upper Arm, and Forearm, are defined based on the anatomical landmark positions following the recommendations of the International Society of Biomechanics (ISB) [47]. In the following equations, body segment Thorax, Torso, Upper Arm and Forearm are defined as λ, μ, η, and ψ, respectively. The origin of a segment is denoted by o. The axes of each coordinate system are denoted by x, y and z.

The origin $o_{\Gamma ,\lambda }$ of the thorax segment is defined as the midpoint between C7 and CLAV. The origin $o_{\Gamma ,\mu }$ of the torso segment is defined as the midpoint of T10 and STRN. The y-axis of thorax coordination system $y_{\Gamma ,\lambda }$ is defined by the unit vector going from the midpoint of T10 and STRN to the midpoint of C7 and CLAV, pointing upwards. The z-axis of the thorax coordinate system $z_{\Gamma ,\lambda }$ is defined by the unit vector perpendicular to the plane defined by T10, C7 and CLAV, pointing laterally. The x-axis of the thorax coordinate system $x_{\Gamma ,\lambda }$ is defined by the unit vector perpendicular to the plane defined by the y-axis and z-axis to create a right-hand coordinate system. The coordinate system of the thorax segment $C_{\Gamma ,\lambda }$ is then constructed by its x, y, and z-axis.

The origin $o_{\Gamma ,\eta }$ of the right upper arm segment is defined by the elbow joint center E, which is the midpoint between EL and EM. The y-axis of the right upper arm segment $y_{\Gamma ,\eta }$ is defined by the unit vector going from the elbow joint center E to shoulder joint center S, which is the center of PSH, ASH and ACR. The z-axis of the right upper arm segment $z_{\Gamma ,\eta }$ is defined as the unit vector perpendicular to the plane formed by the y-axis of the upper arm and the long axis vector of the forearm. The x-axis $x_{\Gamma ,\eta }$ is defined by the y-axis and the z-axis of the right upper arm segment to create a right-hand coordinate system. The coordinate system of the upper arm segment $C_{\Gamma ,\eta }$ is then constructed by x, y and z-axis of the segment.

The origin $o_{\Gamma ,\psi }$ of the right forearm segment coordinate system is defined by the wrist joint center W, which is the midpoint between RS and US. The y-axis of the right forearm segment coordinate system $y_{\Gamma ,\psi }$ is defined by the unit vector from the wrist joint center W to the elbow joint center E, pointing upwards. The x-axis of the right forearm segment coordinate system $x_{\Gamma ,\psi }$ is defined by the unit vector perpendicular to the plane formed by y-axis and the vector from US to RS, pointing anteriorly. The z-axis $z_{\Gamma ,\psi }$ is defined by the unit vector perpendicular to the x and y-axis of right forearm segment, pointing anteriorly. The coordinate system of the forearm segment $C_{\Gamma ,\psi }$ is then constructed by x, y and z-axis of the segment.

The calibrated anatomical systems technique [48] is used to establish the motion of anatomical landmarks relative to the coordinate systems of the upper arm cluster (PUA) or the forearm cluster (DUA). The motion of the upper-limb landmarks could be reconstructed from their constant relative positions to the upper-arm technical coordinate system. For each sampling time frame, the coordinates of each segment with respect to its proximal segment are transformed by a sequence of three rotations following z-x-y order.

The rotation matrix $R_{\Gamma }\left(\lambda ,\eta \right)$ and $R_{\Gamma }\left(\eta ,\psi \right)$ are obtained via their responsive parent coordinate system and child coordination system. Shoulder flexion (+)/extension (−) $\beta_{\mathrm{FE}}$, adduction (+)/abduction (−) $\beta_{\mathrm{AA}}$, internal (+)/external rotation (−) $\beta_{\mathrm{IE}}$, elbow flexion (+)/extension (−) $\beta_{\mathrm{EFE}}$, varus (+)/valgus (−) $\beta_{\mathrm{EVV}}$, internal/external rotation $\beta_{\mathrm{EIE}}$ angles are calculated by solving the multivariable equations.

The UWA upper limb kinematic model Γ is developed using the Vicon Bodybuilder software (Oxford Metrics Group). The reference shoulder angles and elbow flexion/extension angle $k_{\Gamma }=\left[\beta_{FE},\beta_{AA},\beta_{IE},\beta_{EFE}\right]$ are used as a golden standard to train our deep learning refined model for the Kinect v2 system. We use *fourth-order zero-lag Butterworth low-pass* filter with the cut-off frequency of 6 Hz for the UWA model Γ as well as the Kinematic model for Kinect Φ. The cut-off frequency is followed the recommendation from the literature and determined by residual analysis for the upper limb tasks [49].

### 2.3. Long Short-Term Memory Neural Network

We construct a recurrent neural network [41] *N* to refine the upper limb kinematics $k_{\Phi }=\left[\alpha_{FE},\alpha_{AA},\alpha_{IE},\alpha_{EFE}\right]$ calculated by the kinematic model for Kinect v2 (see Section 2.1). In order to reduce the systematic error of Φ, the kinematics $k_{\Gamma }=\left[\beta_{FE},\beta_{AA},\beta_{IE},\beta_{EFE}\right]$ calculated by the UWA model for the 3DMC system (see Section 2.2) is taken as a target. To adapt the neural network, we assume that $k_{\Phi }$ and $k_{\Gamma }$ are normalize into range [0,1].

As shown in Figure 5, our neural network is formed by three long short-term memory (LSTM) layers. The input of our model is a 101-time-step sequence (t=101). The unit of each time-step is a 4-dimensional vector. We use, empirically, 100 neural units in each LSTM cell. The output of the model is also a 101-time-step sequence with 4-dimensional vectors.

To train this model, we let $k_{\Phi }$ be the input of the model. The output is denoted by $k_{\hat{\Phi }}=\left[{\hat{\alpha }}_{FE},{\hat{\alpha }}_{AA},{\hat{\alpha }}_{IE},{\hat{\alpha }}_{EFE}\right]$. We calculate the mean square error between $k_{\hat{\Phi }}$ and $k_{\Gamma }$ as the loss of the model, and then employ an Adam method for optimization [50]. The network is trained with a batch size of 20 and the learning rate is set to 0.006 for 200 epochs. In application, the upper limb kinematics $k_{\Phi }$ calculated by Kinect v2 is taken as the input of the neural network. The output of the neural network, namely $k_{\hat{\Phi }}$, is our refined upper limb kinematics.

## 3. Evaluation

### 3.1. Subjects

We recruited thirteen healthy male university students (age: 25.3 ± 2.5 years old; height: 173.2 ± 4.1 cm; mass: 69.1 ± 6.5 Kg). The participants were absent of any upper limb neuromusculoskeletal problems or medication use that would affect their upper limb functions. The participants were informed about the basic procedure of the experiment before the test. The experimental protocol was approved by the Research Academy of Grand Health’s Ethics Committee at Ningbo University and all participants provided written informed consent.

### 3.2. Experiment Protocol

We used a concurrent validity design to evaluate our deep learning based upper limb functional assessment system using the Kinect v2 sensor. The 3D anatomical position of the upper limb (take the right side as an example) and trunk were recorded concurrently by a Kinect v2 sensor and a 3DMC system with eight high-speed infrared cameras (Vicon, Oxford Metrics Ltd., Oxford, UK). The Kinect v2 sensor and the 3DMC recorded the position of anatomical landmarks with sampling frequencies of around 30 Hz and 100 Hz, respectively. The Kinect sensor was placed on a tripod, 0.8 meters above the ground, and 2 meters in front of the subject [51].

Optical reflective markers were attached to the anatomical landmarks of each individual following the instruction of the UWA upper limb marker set [39]. A static trial was recorded firstly during which the participant was standing in the anatomical position. The elbow and wrist markers were removed during dynamic trials. Four functional tasks, as shown in Figure 6, representing a variety of active daily functional activities [42] and at the same time are important for independent living [52], were performed. The tasks were selected based on previous studies [42,52,53,54,55] after extensive consultation with clinicians. These tasks are also used in assessment scales such as Mallet score, which is commonly used for evaluation of shoulder function [56].

Task 1: Hand to the contralateral shoulder, which represents all activities near contralateral shoulder such as washing axilla or zip up a jacket. Subjects started with the arm in the anatomical position with their hand handing beside their body in a relaxed neutral position and end up with the hand touched the contralateral shoulder (see Figure 6, left); 

Task 2: Hand to mouth or drinking, which represents activities such as eating and reaching the face. It begins with the same starting point, and ends when the hand reached subject’s mouth (see Figure 6, middle-left); 

Task 3: Combing hair, which represents activities such as reaching the (back of the) head and washing hair. Subjects were instructed to move their hand to the back of their head (see Figure 6, middle-right);

Task 4: Hand to back pocket, which represents reaching the back and perineal care. It begins with the same starting point and ends when the hand placed on the back pocket (see Figure 6, right).

### 3.3. Leave One Subject Out Cross-Validation

We firstly calculate upper limb kinematics $k_{\Phi }$ and $k_{\Gamma }$ using upper limb kinematic model for Kinect v2 system Φ and the UWA kinematic model Γ for the reference 3DMC system, respectively. For all four functional tasks, the joint angles are resampled to 101-time steps. Joint angles are represented as 0–100% across the time domain, with 0% being the initial and 100% being the finish. Next, we use a leave one subject out cross-validation (LOOCV) (see Figure 7) to evaluate the performance of our proposed deep learning refined upper limb functional assessment model $\hat{\Phi }$ using Kinect v2 sensor.

Using the LOOCV protocol, the kinematic data $k_{\Phi }$ and $k_{\Gamma }$ are partitioned into training data and test data. Assuming that we have n subjects, the validation process iterates n times. For each iteration, kinematic data of the left-out subject is set as the testing data and the kinematics of the remaining subjects is set as the training data. The testing data include one 3D matrix, which are the shoulder and elbow joint angles of the left-out subject calculated via the kinematic model for Kinect v2 system Φ. The training data from the remaining subjects are consist of two 3D matrices, the upper limb joint angles calculated via model Φ, regarded as the input data of the deep learning refined kinematic model $\hat{\Phi }$ for Kinect v2, and the reference UWA kinematic model Γ for the 3DMC system, regarded as the target data of the model $\hat{\Phi }$. Our deep learning refined kinematic model $\hat{\Phi }$ explores the nonlinear relationship between the upper limb kinematics via the kinematics model for Kinect $k_{\Phi }$ and those angles via the UWA model using the 3DMC system $k_{\Gamma }$. Such a model can reduce the systematic error of the Kinect system. 

### 3.4. Performance Evaluation and Statistical Analysis of the Deep Learning Refined Kinematic Model

The performance of our developed model $\hat{\Phi }$ is evaluated based on the test data, using the upper limb kinematics calculated via the model Γ for the 3DMC system as the ground truth. The coefficient of multiple correlation (CMC) values and root mean squared errors (RMSE) are calculated between upper limb kinematic waveforms $k_{\Phi }$ and $k_{\Gamma }$ as well as between $k_{\hat{\Phi }}$ and $k_{\Gamma }$ in each application session.

The CMC values are used to evaluate the similarity and repeatability of the upper limb joint angle trajectories between $k_{\Phi }$ and the $k_{\Gamma }$ as well as the similarities between $k_{\hat{\Phi }}$ and $k_{\Gamma }$. The CMCs are calculated following Kadaba’s approach [57]. The CMC values are explained as excellent similarity (0.95–1), very good similarity (0.85–0.94), good similarity (0.75–0.84), moderate similarity (0.6–0.74) and poor similarity (0–0.59) [58]. The RMSE values are employed to evaluate mean errors between the upper limb angle waveforms $k_{\Phi }$ and the $k_{\Gamma }$ as well as errors between $k_{\hat{\Phi }}$ and $k_{\Gamma }$ across all functional tasks.

Range of motion (ROM) values and the joint angle at the point of target achieved (PTA) via the kinematic model Φ, our deep learning refined kinematic model $\hat{\Phi }$ for Kinect v2 system and the UWA kinematic model Γ for the 3DMC system are calculated and extracted. Both ROM and PTA data are extracted from the test data in the application process. The normality of all ROM and PTA values are tested by the Shapiro-Wilks test (p > 0.05). A paired sample t-test is used for the parameters which are normally distributed; the Wilcoxon Signed Ranks Test is used for those who are not. Bland-Altman analysis with 95% limits of agreement (LoA) is performed to assess the agreements between the ROMs and the PTAs via model Φ and model Γ as well as the agreements via model $\hat{\Phi }$ and model Γ. The CMC and RMSE are analyzed using Matlab 2019a, and the rest statistical analysis is carried out using SPSS 25.0.

## 4. Results

### 4.1. Joint Kinematic Waveforms Validity

The kinematic waveforms of the chosen representative upper limb functional tasks via the kinematic model Φ and our deep learning refined kinematic model $\hat{\Phi }$ for the Kinect v2 system are presented in Figure 8, Figure 9, Figure 10 and Figure 11 by means of average angles from the testing data. Joint angles via the UWA kinematic model Γ for the 3DMC system are presented in Figure 8, Figure 9, Figure 10 and Figure 11 as the golden standard. The CMC values between $k_{\Phi }$ and $k_{\Gamma }$ as well as between $k_{\hat{\Phi }}$ and $k_{\Gamma }$ are presented in Table 1. The RMSE values are presented in Table 2.

Our model $\hat{\Phi }$ significantly improves the waveform similarity (see Table 1) and decreases the RMSE (see Table 2) in comparison with the model Φ for Kinect v2 for almost all upper limb joint angles during all investigated functional tasks (p < 0.05). For the angles calculated with model Φ, very good similarities (CMC = 0.85–0.94) are only observed in shoulder flexion/extension angles during Task 1 and Task 4 with the mean CMCs of 0.87 and 0.92 and in elbow flexion/extension angles during Task 1 and Task 2 with the mean CMCs of 0.93 and 0.92. Good similarities (CMC = 0.75–0.84) are only observed in shoulder flexion/extension angle during Task 3 (CMC = 0.77), in shoulder adduction/abduction angle (CMC = 0.84) and shoulder internal/external rotation angle (CMC = 0.81) during Task 1 and in elbow flexion/extension angle during Task 2 and Task 3 (CMC = 0.83 for both tasks). For the rest upper limb joint angles during all chosen functional tasks, angles calculated using model Φ show poor to moderate waveform similarities in comparison with the reference joint angles (CMC = 0.55–0.74).

The RMSEs between $k_{\Phi }$ and the $k_{\Gamma }$ as well as the RMSEs between $k_{\hat{\Phi }}$ and $k_{\Gamma }$ also demonstrate the promising ability of our deep learning refined kinematic model $\hat{\Phi }$ in increasing upper limb joint kinematic accuracy using Kinect v2. The RMSEs are both plane-dependent and task-dependent. Our model $\hat{\Phi }$ decreases the RMSEs with much lower mean values and standard deviations for all degrees of freedom under all functional tasks in comparison model Φ. The RMSEs via our model $\hat{\Phi }$ are significantly smaller than those via model Φ (p < 0.05) except for shoulder flexion/extension angles during the hand to back pocket task. For shoulder flexion/extension angle during the hand to back pocket task, despite the RMSEs via our model $\hat{\Phi }$ and via model Φ do not reach significant difference, the RMSEs via both models are all relatively small. Our model $\hat{\Phi }$ yields lower RMSEs. Taking the combing hair task as an example, the RMSEs drop from 41.73° ± 8.19° to 11.50° ± 7.25° for shoulder flexion/extension angles, from 11.91° ± 4.61° to 5.14° ± 1.83° for shoulder adduction/abduction angles, from 31.45° ± 6.89° to 8.59° ± 2.91° for shoulder internal/external rotation angles and from 25.83° ± 3.45° to 6.96° ± 2.92° for elbow flexion/extension angles after using model $\hat{\Phi }$ instead of model Φ.

Using our deep learning refined kinematic model $\hat{\Phi }$, shoulder and elbow flexion/extension angles during all four functional tasks show excellent similarities between $k_{\hat{\Phi }}$ and $k_{\Gamma }$ with the mean CMC of 0.95–0.99 except for slightly lower similarities during Task 4 (mean CMC = 0.94 and 0.93 for shoulder and elbow joint respectively). The shoulder internal/external rotation angles show excellent similarity (mean CMC = 0.98) during Task 1, very good similarity (mean CMC = 0.89) during Task 3 and good similarity during Task 2 and Task 4 (mean CMC = 0.75 for both tasks). For shoulder adduction/abduction angles, excellent similarity (mean CMC = 0.97), very good similarity (mean CMC = 0.88) and good similarity (mean CMC = 0.79) are observed in Task 3, Task 1 and Task 4, respectively. The lowest similarity is found for the shoulder adduction/abduction angles during the drinking water task with the mean CMC of 0.72.

### 4.2. Joint Kinematic Variables Validity

The joint angles at the point of target achieved (PTA) and the range of motion (ROM) during the upper limb functional tasks via the kinematic model Φ and our deep learning refined kinematic model $\hat{\Phi }$ for the Kinect v2 system as well as via the UWA kinematic model Γ for the 3DMC system are presented in Table 3 and Table 4, by means of mean and standard deviation values (± SD). Differences and statistical significance of PTAs via model Φ and model $\hat{\Phi }$ in comparison with the PTAs via model Γ are given in Table 3; whereas the absolute errors and statistical significance of ROMs are given in Table 4. The Bland-Altman plots for all PTAs and ROMs are presented in Figure 12, Figure 13, Figure 14 and Figure 15.

The PTAs via the model Φ are all reached significant difference in comparison with those via the refence model Γ (p<0.05) except the shoulder flexion/extension angle during the hand to back pocket task and the shoulder adduction/abduction angle during the combing hair task. In contrast, there is no significant difference in all PTAs via our model $\hat{\Phi }$ and the references except for those of the elbow flexion/extension angles during the hand to back pocket task (p=0.045). Although statistical significance exists in the PTAs of elbow flexion/extension angle during the combing hair task ($\hat{\Phi }$ = 146.29° ± 4.21°, Γ = 144.56° ± 3.49°), our model $\hat{\Phi }$ does reduce the absolute error of the PTA via model Φ from (Φ−Γ) −32.39° ± 4.16° to ($\hat{\Phi }-\Gamma$) 1.73° ± 2.79°. By employing our model $\hat{\Phi }$, the PTAs discrepancy of shoulder and elbow flexion/extension angles between the Kinect v2 and the 3DMC system drop from 45.61° ± 10.30° and -32.39° ± 4.16° (Φ−Γ) to 1.79° ± 11.78° and 1.73° ± 2.79° ($\hat{\Phi }-\Gamma$) respectively during the combing hair task.

There is significant difference in all ROMs via model Φ and the references (p<0.05); whereas there is no significant difference between the ROMs via our model $\hat{\Phi }$ and the references for all investigated upper limb joint angles. The greatest improvement occurs in the ROMs of the shoulder flexion/extension angles during the combing hair task, in which the absolute error between the Kinect v2 and the 3DMC system drop from 40.44° ± 9.88° (Φ−Γ) to 2.95° ± 10.88° ($\hat{\Phi }-\Gamma$).

## 5. Discussion

Our study developed a novel deep learning refined kinematic model for 3D upper limb kinematic assessment using a single Kinect v2 sensor. Our refined model $\hat{\Phi }$ is in good agreement with the 3DMC system and is far more accurate than the traditional kinematic model using the same Kinect v2 sensor for upper limb waveforms, joint angles at the point of target achieved (PTA), and the range of motions (ROM) across all functional tasks. Using our deep learning-based model, the Kinect v2 could measure shoulder and elbow flexion/extension waveforms with mean CMCs >0.93 for all investigated tasks, shoulder adduction/abduction, and internal/external rotation waveforms with mean CMCs >0.8 for most of the tasks. The mean deviations of angles at the PTA and ROM are under 5° for all investigated joint angles during all investigated functional tasks. In clinic application, generally less than 2° is considered acceptable, an error between 2°–5° may also be acceptable with appropriate interpretation [59,60]. Thus, the performance of our deep learning refined kinematic model using a single Kinect v2 sensor is promising as an upper limb functional assessment system.

The results agree with other studies on similar upper limb functional tasks [42]. During the combing hair task, at the maximum elevation, the mean elbow flexion via our model $\hat{\Phi }$ is 146°. This is in agreement with results of van Andel et al. [42], Magermans et al. [52] and Morrey et al. [61], who find average elbow flexion angles of 122°, 136° and 100°, respectively. Andel et al. [42] find that the shoulder flexion angles reach nearly 100° in the combing hair tasks and stay under 70° during the other tasks. This is also the same case in our study via our model $\hat{\Phi }$ using a Kinect v2 sensor. Shoulder flexion angles are around 108° during the hair combing task and remain under 60° during the hand to contralateral shoulder and the hand to mouth task. The hand to mouth task does not require the full ROM of all joints and the most important joint angle is elbow flexion [52]. The mean elbow flexion is 112° via our model $\hat{\Phi }$, which is consistent with Magermans et al.’s research with the elbow flexion of 117° [52].

The systematic errors of the proposed Kinect-based upper limb assessment system include errors due to the inaccurate depth measurement and the motion artifact of moving objects [10]. Kinect v2 measures the depth information based on the Time of Flight (ToF) technique. The ToF measures the time that “light emitted by an illumination unit requires to travel to an object and back to the sensor array”. Kinect v2 utilizes the Continuous Wave (CW) Intensity Modulation approach, which requires several correlated images for calculation of each depth image. The distance calculated based on the mixing of correlated images requires approximation on the CW algorithm and causes systematic error in depth measurement. Recording and processing the correlated images are also both affected by moving objects, which lead to inaccurate depth measurement at object boundaries [10].

The systematic errors also include error due to the kinematic modeling. In both kinematic models, the shoulder joint angles are considered as humerus coordinate rotations relative to the thorax coordinate systems. The kinematic models developed for the Kinect v2 sensor and the model used for the 3DMC system are followed the same recommendation on the definition of joint coordinate systems of trunk, shoulder, and elbow joint proposed by the International Society of Biomechanics [47,62]. The second option of humerus coordinate system is used for both systems [47], in which the z-axis of the humerus coordinate system is perpendicular to the plane formed by the vector from the elbow joint center to the shoulder joint center and the vector from the wrist joint center to the elbow joint center. For the UWA model, the thorax segment is defined by the 7th cervical vertebra, the 10th thoracic vertebra, the sternoclavicular notch and the xyphoid process of the sternum. Because of the limited ability of skeletal joint tracking in the Kinect based system, the thorax coordinate system is defined by Kinect Skeleton landmarks of both trunk segment and shoulder joints (i.e., SpineShoulder, SpineMid, ShoulderLeft and ShoulderRight). Thus, tasks with large clavicle movements such as combing hair have great deviations in shoulder kinematic assessment. In our study, the shoulder joint angles during the combing hair task yield the largest root-mean-squared errors using our deep learning refined model in comparison with the golden standard system.

From Figure 8, Figure 9, Figure 10 and Figure 11, it can be seen that the systematic error of the Kinect based system is highly nonlinear. The LSTM network we employed is the state-of-the-art recurrent neural network, which is good at modeling the nonlinear relationship for time series data. Our deep learning based algorithm yields better results to the linear regression algorithm [63] in refining joint angles using a single Kinect sensor. In assessing shoulder joint angles during the computer-using task, only shoulder adduction/abduction is improved after the linear regression refinement [63]. As the measurement error is positively correlated with the magnitude of that joint angle [63], the measurement error is presented with its ROM. After applying the linear regression calibration, the mean RMSE of the shoulder adduction/abduction angle are decreased from 14.8° and 9.1° for the right and left shoulder respectively to 7.5°, during which the ROM of the angle is under 20°. While using our deep learning refined kinematic model $\hat{\Phi }$, all upper limb joint angles, including shoulder flexion/extension, shoulder adduction/abduction, shoulder internal/external rotation, and elbow flexion/extension, are significantly improved during all functional tasks. Notably, the mean RMSEs of shoulder adduction/abduction angles are decreased to around 3° for task 1, task 2 and task 4 and to around 5°for task 3 with the mean ROMs of 12.97° to 47.36°.

Previous studies reveal that Kinect v2 with the automated body tracking algorithm is also not suitable to assess lower-body kinematics. The deviation of hip flexion during the swing phase is more than 30° during walking [15]. The limits of agreement (LoA) between the Kinect v2 sensor and the 3DMC system are 28°, 46° for peak knee flexion angle at a self-selected walking speed [15], 7°, 25° for trunk anterior-posterior flexion [16]. Average errors of 24°, 26° are observed for the right and left peak knee flexion angles during squatting [19]. 

Timmi et al. [37] employed custom-made colored markers placed on bony prominences near the hip, knee, and ankle. The marker tracking approach improves the knee angle measurement with the LOA of −1.8° and 1.7° for flexion and −2.9°, 1.7° for adduction during fast walking. Compared with gait analysis and static posture assessment, motion analysis of the upper limb using Kinect sensors is far more challenging. Upper limb functional activities show a larger variation in the healthy population and a higher number of degrees of freedom in the upper limb. The upper limb, especially the shoulder joint, has a very large working range, comparing to the lower extremity. Furthermore, the upper limb joints are easy to be occluded by each other. The marker-tracking methodology may not be suitable for the Kinect based system in assessing upper limb kinematics.

## 6. Conclusions

We have developed a novel deep learning refined kinematic model for upper limb functional assessment using a single Kinect v2 sensor. The system demonstrates good kinematic accuracy in comparison with a standard marker-based 3D motion capture system during performing upper limb functional tasks, suggesting that such a single-Kinect-based kinematic assessment system has great potential to be used as an alternative of the traditional marker-based 3D motion capture system. Such a low-cost, easy to use system with good accuracy will help small rehabilitation clinics or meet the need for rehabilitation at home.

## Figures and Tables

**Figure 1 sensors-20-01903-f001:**
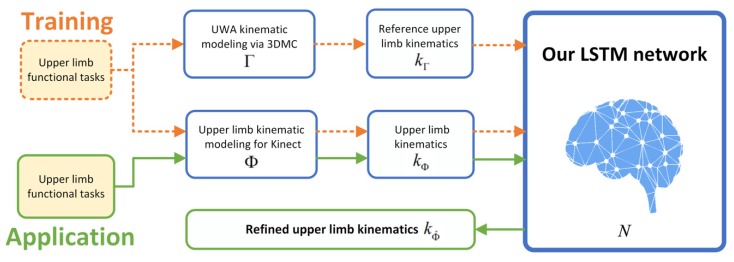
The architecture of our deep learning refined kinematic model for Kinect v2.

**Figure 2 sensors-20-01903-f002:**
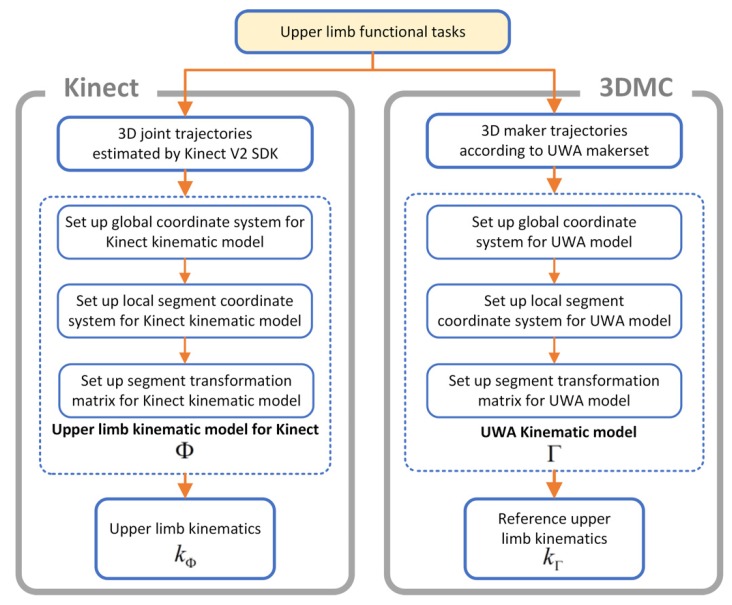
The kinematic models of the Kinect v2 system and the 3D motion capture system.

**Figure 3 sensors-20-01903-f003:**
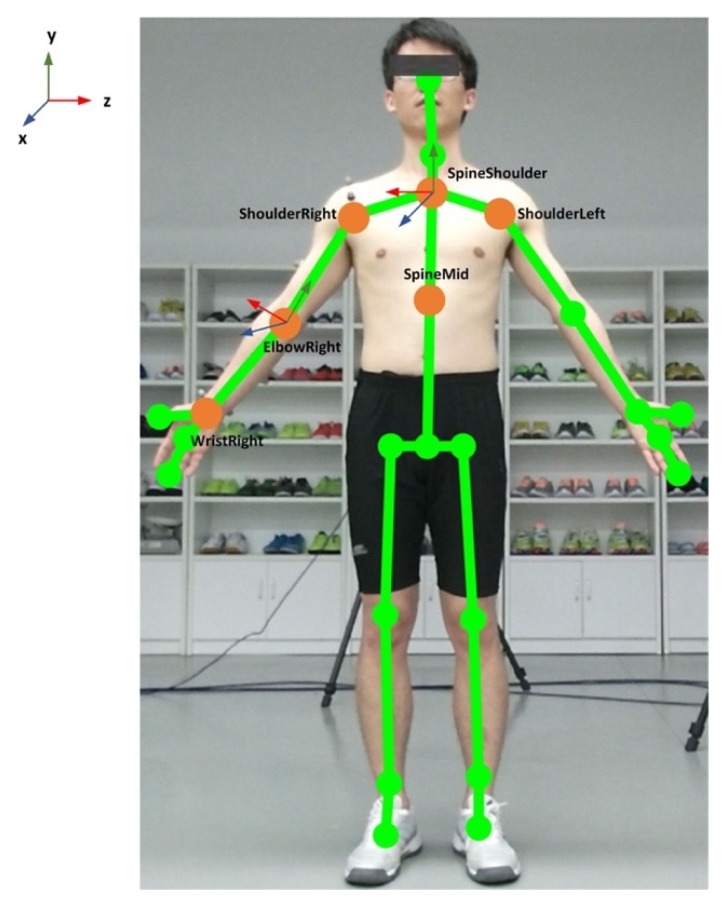
Illustration of the Kinect skeleton joints and the anatomical coordination system.

**Figure 4 sensors-20-01903-f004:**
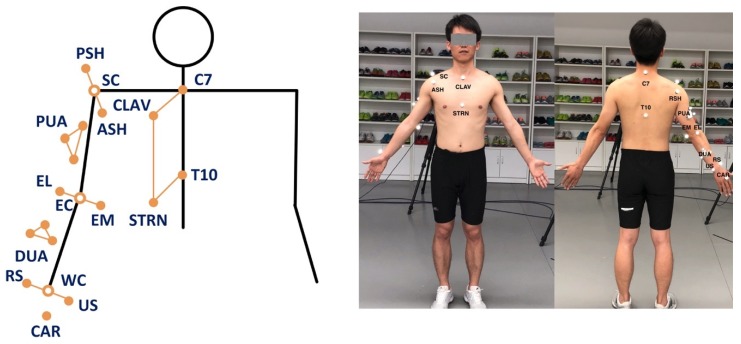
Marker set for the 3DMC system. **Left**: Arrangement of the UWA upper limb marker set. **Right**: A participant with the attached markers.

**Figure 5 sensors-20-01903-f005:**
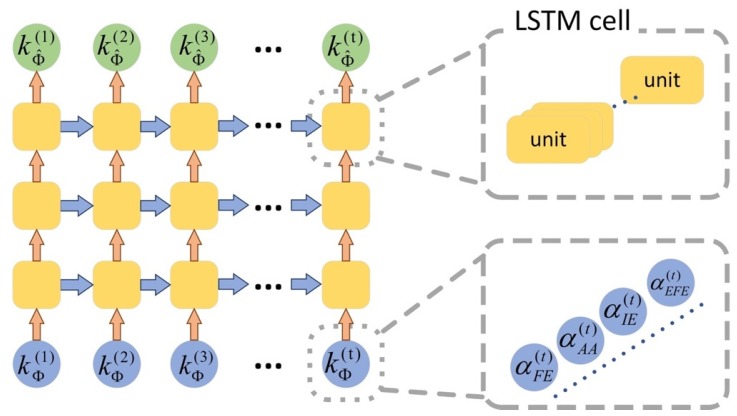
Architecture of our LSTM neural network for upper limb kinematics refinement.

**Figure 6 sensors-20-01903-f006:**
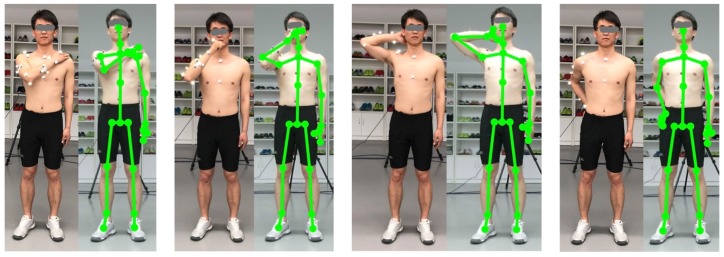
Four upper limb functional tasks evaluated in our study. **Left**: Hand to the contralateral shoulder. Middle-left: Hand to mouth or drinking. Middle-right: combing hair. **Right**: Hand to back pocket.

**Figure 7 sensors-20-01903-f007:**
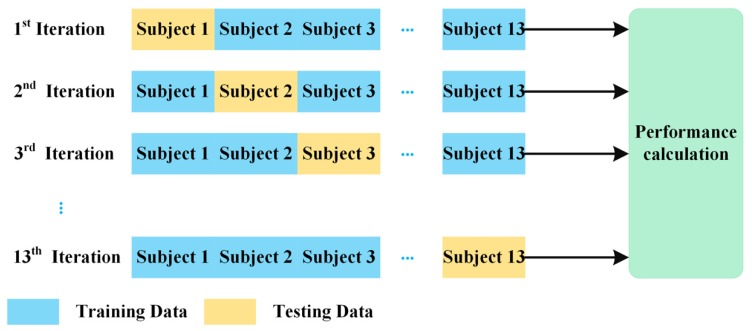
The protocol of the leave one subject out cross validation (LOOCV).

**Figure 8 sensors-20-01903-f008:**
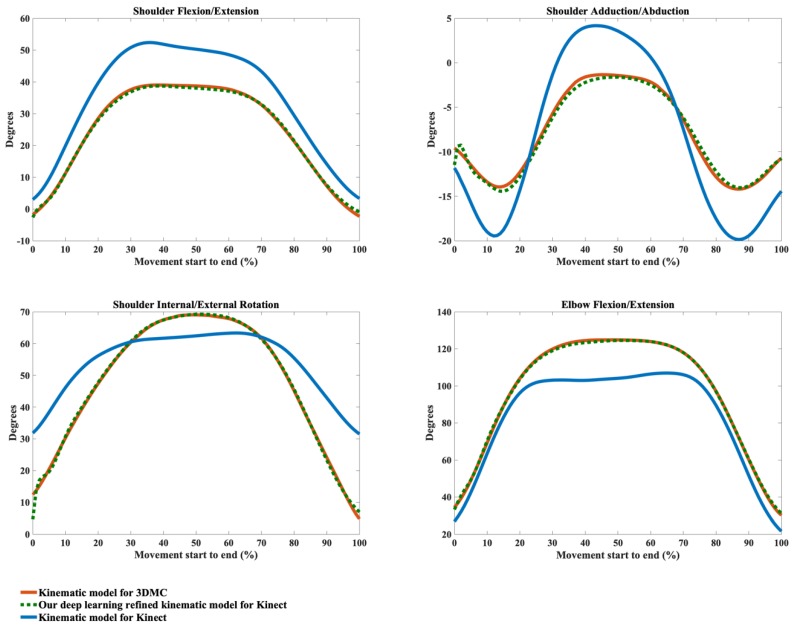
Joint angles during the hand to contra lateral shoulder task calculated via the kinematic model for 3DMC (orange solid line), our deep learning refined kinematic model for Kinect (green dashed line) and the kinematic model for Kinect (blue solid line). The joint angles include shoulder flexion (+)/extension (−), shoulder adduction (+)/abduction (−), shoulder internal rotation (+)/external rotation (−), and elbow flexion (+)/extension (−).

**Figure 9 sensors-20-01903-f009:**
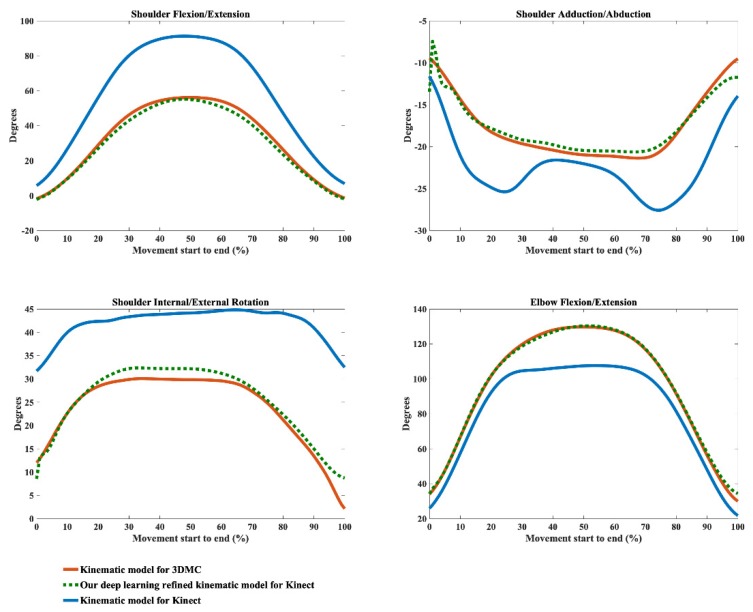
Joint angles during the hand to mouth task calculated via the kinematic model for 3DMC (orange solid line), our deep learning refined kinematic model for Kinect (green dashed line) and the kinematic model for Kinect (blue solid line). The joint angles include shoulder flexion (+)/extension (−), shoulder adduction (+)/abduction (−), shoulder internal rotation (+)/external rotation (−), and elbow flexion (+)/extension (−).

**Figure 10 sensors-20-01903-f010:**
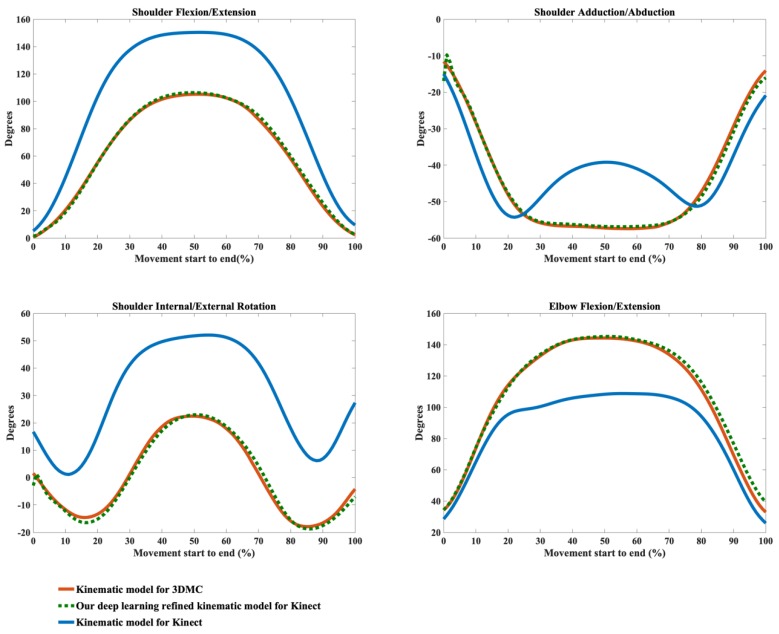
Joint angles during the combing hair task calculated via the kinematic model for 3DMC (orange solid line), our deep learning refined kinematic model for Kinect (green dashed line) and the kinematic model for Kinect (blue solid line). The joint angles include shoulder flexion (+)/extension (−), shoulder adduction (+)/abduction (−), shoulder internal rotation (+)/external rotation (−), and elbow flexion (+)/extension (−).

**Figure 11 sensors-20-01903-f011:**
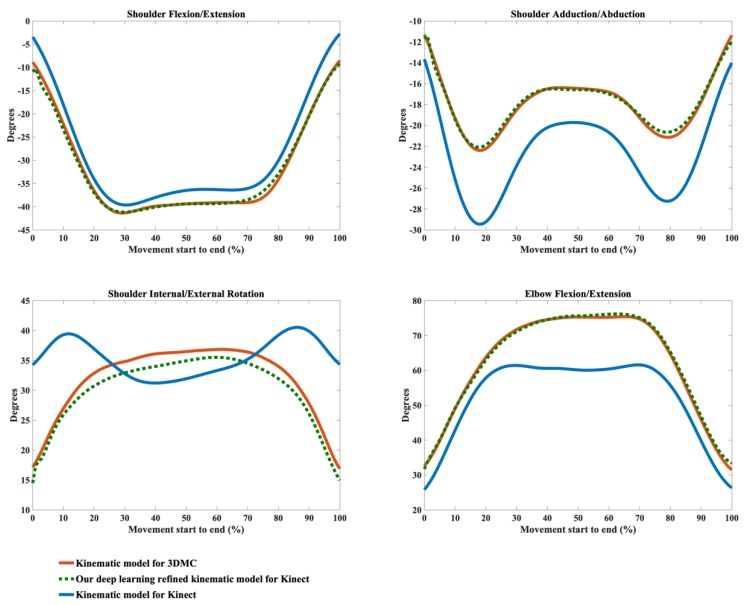
Joint angles during the hand to back pocket task calculated via the kinematic model for 3DMC (orange solid line), our deep learning refined kinematic model for Kinect (green dashed line) and the kinematic model for Kinect (blue solid line). The joint angles include shoulder flexion (+)/extension (−), shoulder adduction (+)/abduction (−), shoulder internal rotation (+)/external rotation (−), and elbow flexion (+)/extension (−).

**Figure 12 sensors-20-01903-f012:**
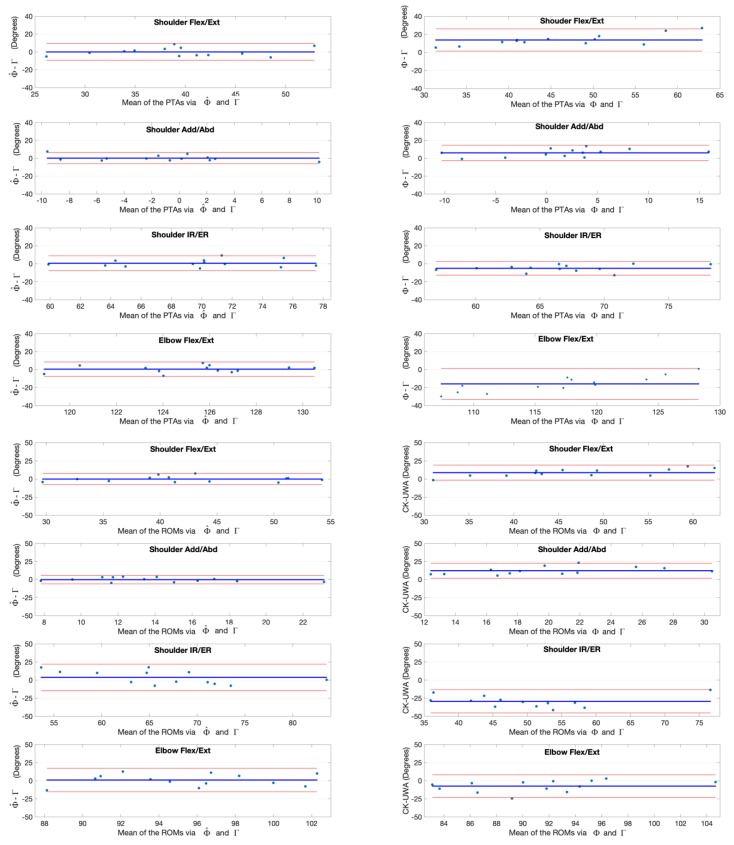
Bland-Altman plots with 95% limits of agreement for joint kinematic parameters during the hand to contralateral shoulder task. X axes represents the angle means of two systems and the Y axes represents the mean of differences. The red line (middle one) represents the reference line at mean, and the two dashed lines represent the upper and lower limit of agreement. The upper four rows are the angles at the point of target achieved (PTA) and the lower four rows are the range of motion (ROM) values. Plots of the left column are measurement differences between our deep learning refined kinematic model $\hat{\Phi }$ for Kinect and the UWA kinematic model Γ for the 3DMC. Plots of the right column are measurement differences between the kinematic model Φ for Kinect and the UWA kinematic model for the 3DMC Γ.

**Figure 13 sensors-20-01903-f013:**
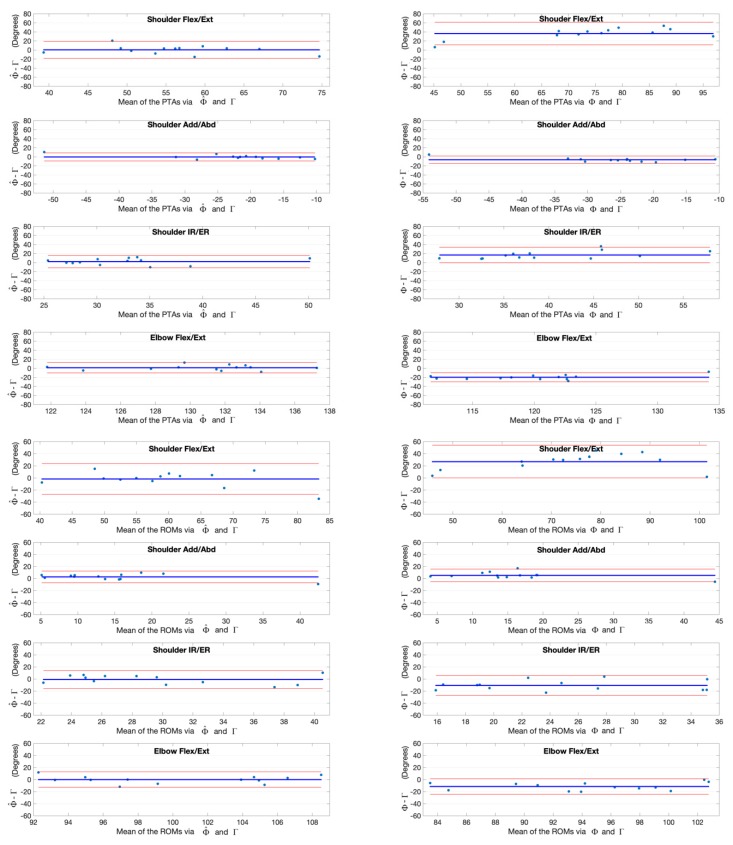
Bland-Altman plots with 95% limits of agreement for joint kinematic parameters during the hand to mouth task. X axes represents the angle means of two systems and the Y axes represents the mean of differences. The red line (middle one) represents the reference line at mean, and the two dashed lines represent the upper and lower limit of agreement. The upper four rows are the angles at the point of target achieved (PTA) and the lower four rows are the range of motion (ROM) values. Plots of the left column are measurement differences between our deep learning refined kinematic model $\hat{\Phi }$ for Kinect and the UWA kinematic model Γ for the 3DMC. Plots of the right column are measurement differences between the kinematic model Φ for Kinect and the UWA kinematic model for the 3DMC Γ.

**Figure 14 sensors-20-01903-f014:**
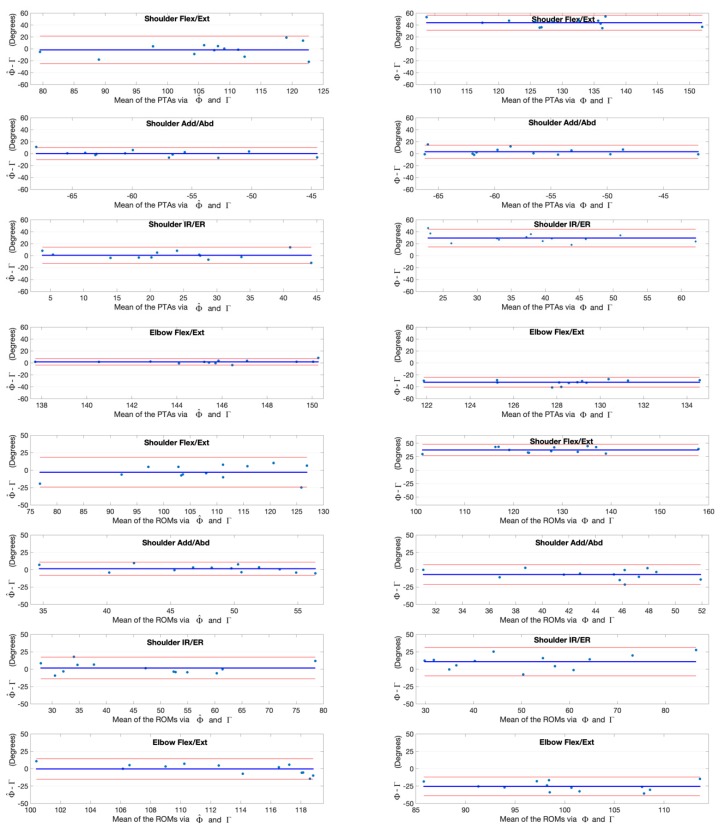
Bland-Altman plots with 95% limits of agreement for joint kinematic parameters during the combing hair task. X axes represents the angle means of two systems and the Y axes represents the mean of differences. The red line (middle one) represents the reference line at mean, and the two dashed lines represent the upper and lower limit of agreement. The upper four rows are the angles at the point of target achieved (PTA) and the lower four rows are the range of motion (ROM) values. Plots of the left column are measurement differences between our deep learning refined kinematic model $\hat{\Phi }$ for Kinect and the UWA kinematic model Γ for the 3DMC. Plots of the right column are measurement differences between the kinematic model Φ for Kinect and the UWA kinematic model for the 3DMC Γ.

**Figure 15 sensors-20-01903-f015:**
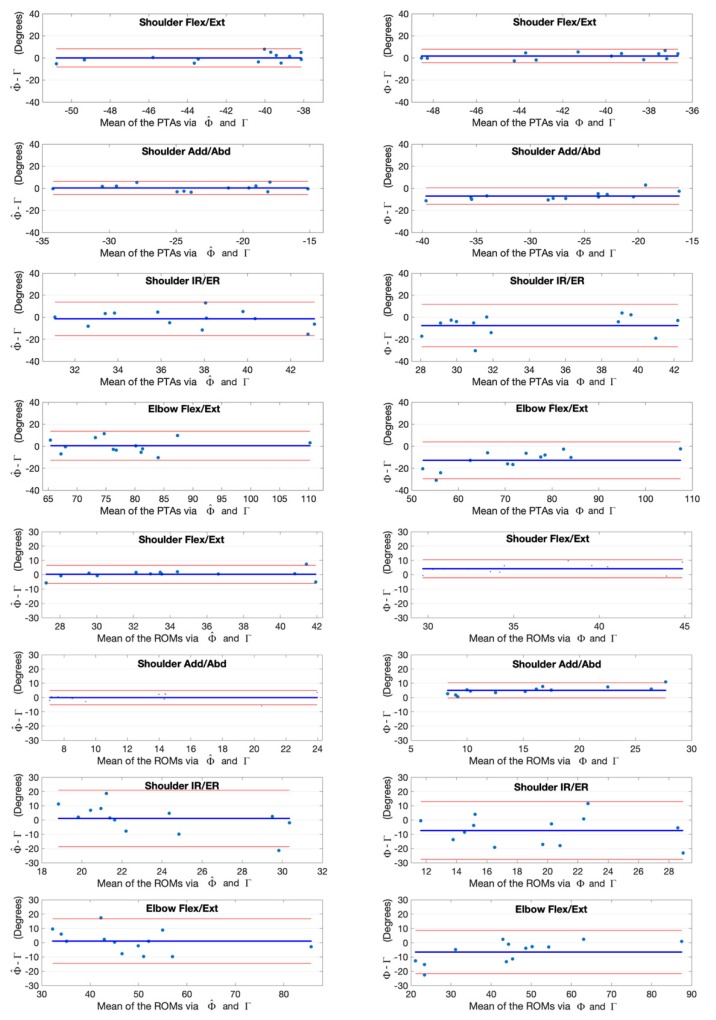
Bland-Altman plots with 95% limits of agreement for joint kinematic parameters during the hand to back pocket task. X axes represents the angle means of two systems and the Y axes represents the mean of differences. The red line (middle one) represents the reference line at mean, and the two dashed lines represent the upper and lower limit of agreement. The upper four rows are the angles at the point of target achieved (PTA) and the lower four rows are the range of motion (ROM) values. Plots of the left column are measurement differences between our deep learning refined kinematic model for Kinect and the UWA kinematic model for the 3DMC. Plots of the left column are measurement differences between our deep learning refined kinematic model $\hat{\Phi }$ for Kinect and the UWA kinematic model Γ for the 3DMC. Plots of the right column are measurement differences between the kinematic model Φ for Kinect and the UWA kinematic model for the 3DMC Γ.

**Table 1 sensors-20-01903-t001:** The coefficient of multiple correlation (CMC) (SD) between the joint angles via kinematic model Φ for Kinect and the angles via 3DMC as well as between angles via our deep learning refined kinematic model $\hat{\Phi }$ for Kinect and the angles via 3DMC.

		Task 1	Task 2	Task 3	Task 4
**Shoulder Flexion** /**Extension**	Φ	0.87 (0.99)	0.73 (0.13)	0.77 (0.13)	0.92 (0.09)
	$\hat{\Phi }$	0.97 (0.20)	0.95 (0.06)	0.97 (0.03)	0.94 (0.08)
	P value	0.005	0.005 ^a^	0.001 ^a^	0.249 ^a^
**Shoulder Adduction** /**Abduction**	Φ	0.84 (0.10)	0.55 (0.27)	0.74 (0.18)	0.60 (0.22)
	$\hat{\Phi }$	0.88 (0.16)	0.72 (0.21)	0.97 (0.02)	0.79 (0.17)
	P value	0.133 ^a^	0.135	0.001	0.016
**Shoulder Internal** /**External Rotation**	Φ	0.81 (0.10)	0.64 (0.23)	0.83 (0.07)	0.65 (0.17)
	$\hat{\Phi }$	0.98 (0.21)	0.75 (0.21)	0.89 (0.11)	0.75 (0.17)
	P value	0.001 ^a^	0.281	0.002 ^a^	0.133
**Elbow Flexion** /**Extension**	Φ	0.93 (0.53)	0.92 (0.07)	0.83 (0.07)	0.83 (0.20)
	$\hat{\Phi }$	0.99 (0.00)	0.99 (0.01)	0.99 (0.01)	0.93 (0.06)
	P value	0.001	0.003 ^a^	0.001 ^a^	0.055 ^a^

Superscript “a” Parameter is not normally distributed; the Wilcoxon Signed Ranks Test is used.

**Table 2 sensors-20-01903-t002:** The root mean squared error (RMSE) (SD) between the joint angles via kinematic model Φ for Kinect and the angles via 3DMC as well as between angles via our deep learning refined kinematic model $\hat{\Phi }$ for Kinect and the angles via 3DMC.

		Task 1	Task 2	Task 3	Task 4
**Shoulder Flexion** **/Extension**	Φ	10.54 (4.65)	27.49 (11.38)	41.73 (8.19)	5.44 (2.91)
	$\hat{\Phi }$	4.61 (1.75)	8.18 (5.35)	11.50 (7.25)	4.39 (2.54)
	P value	0.001	0.003	0.000	0.256
**Shoulder Adduction** **/Abduction**	Φ	5.53 (1.81)	6.42 (2.41)	11.91 (4.61)	6.00 (2.16)
	$\hat{\Phi }$	2.90 (1.42)	3.88 (2.59)	5.14 (1.83)	3.04 (1.72)
	P value	0.004	0.019	0.000	0.000
**Shoulder Internal** **/External Rotation**	Φ	12.80 (3.22)	19.39 (6.76)	31.45 (6.89)	11.34 (3.44)
	$\hat{\Phi }$	5.60 (2.00)	7.15 (2.30)	8.59 (2.91)	6.16 (3.46)
	P value	0.000	0.000	0.000	0.001
**Elbow Flexion** **/Extension**	Φ	14.96 (5.90)	15.83 (4.90)	25.83 (3.45)	11.43 (7.10)
	$\hat{\Phi }$	5.56 (1.18)	7.74 (2.21)	6.96 (2.92)	6.53 (2.41)
	P value	0.000	0.000	0.000	0.028 ^a^

Superscript “a” Parameter is not normally distributed; the Wilcoxon Signed Ranks Test is used.

**Table 3 sensors-20-01903-t003:** The joint angle at the point of target achieved (PTA) with mean and standard deviation (SD) values calculated via the kinematic model Φ for Kinect v2, our deep learning refined kinematic model $\hat{\Phi }$ for Kinect v2 and the UWA model Γ for the 3DMC system. Φ−Γ represents the discrepancy between the PTAs via the model Φ and the reference model Γ. $\hat{\Phi }-\Gamma$ represents the differences between the PTAs via the model $\hat{\Phi }$ and the reference model Γ.

		Task 1	Task 2	Task 3	Task 4
**Shoulder Flexion/Extension (Degrees)**	Φ	53.05 (12.02)	92.37 (20.33)	151.50 (9.96)	−40.30 (4.96)
	$\hat{\Phi }$	39.37 (7.22)	56.14 (11.29)	107.68 (12.24)	−42.04 (3.77)
	Γ	39.35 (7.95)	56.37 (8.83)	105.89 (15.02)	−42.06 (5.56)
	Φ−Γ	13.70 (6.39)	35.99 (15.59)	45.61 (10.30)	1.75 (4.27)
	$\hat{\Phi }-\Gamma$	0.02 (4.79)	−0.24 (9.52)	1.79 (11.78)	0.01 (4.21)
	$p_{\Phi ,\Gamma }$	0.000	0.000 ^a^	0.000	0.163
	$p_{\hat{\Phi },\Gamma }$	0.987	0.930	0.594	0.990
**Shoulder Adduction/Abduction (Degrees)**	Φ	4.74 (7.98)	−29.41 (9.11)	−55.39 (7.03)	−30.72 (8.56)
	$\hat{\Phi }$	−1.19 (4.82)	−23.20 (8.86)	−58.51 (5.68)	−23.39 (5.83)
	Γ	−1.28 (6.15)	−22.78 (12.03)	−58.49 (8.30)	−23.72 (6.01)
	Φ−Γ	6.02 (4.38)	−6.63 (4.26)	3.10 (5.60)	−7.00 (3.82)
	$\hat{\Phi }-\Gamma$	0.10 (3.22)	−0.42 (4.57)	−0.02 (5.21)	0.33 (3.02)
	$p_{\Phi ,\Gamma }$	0.000	0.002 ^a^	0.069	0.000
	$p_{\hat{\Phi },\Gamma }$	0.913	0.382 ^a^	0.988	0.700
**Shoulder Internal/External Rotation (Degrees)**	Φ	64.15 (6.30)	48.43 (11.38)	52.93 (10.21)	30.28 (7.98)
	$\hat{\Phi }$	69.76 (5.76)	33.84 (7.72)	24.04 (11.88)	36.46 (4.55)
	Γ	69.22 (5.30)	31.82 (6.86)	23.53 (13.20)	37.89 (6.16)
	Φ−Γ	−5.08 (3.94)	16.61 (8.69)	29.40 (7.55)	−7.61 (9.82)
	$\hat{\Phi }-\Gamma$	0.53 (4.20)	2.02 (7.06)	0.51 (6.97)	−1.43 (7.76)
	$p_{\Phi ,\Gamma }$	0.001	0.000 ^a^	0.000	0.706
	$p_{\hat{\Phi },\Gamma }$	0.655	0.320 ^a^	0.795	0.517
**Elbow Flexion/Extension (Degrees)**	Φ	109.09 (10.75)	110.18 (7.49)	112.17 (3.96)	65.86 (18.30)
	$\hat{\Phi }$	125.47 (4.18)	131.26 (5.16)	146.29 (4.21)	79.11 (12.15)
	Γ	125.06 (3.42)	130.05 (5.07)	144.56 (3.49)	78.66 (11.85)
	Φ−Γ	−15.97 (8.83)	−19.87 (5.17)	−32.39 (4.16)	−12.80 (8.54)
	$\hat{\Phi }-\Gamma$	0.41 (4.07)	1.21 (5.78)	1.73 (2.79)	0.44 (6.70)
	$p_{\Phi ,\Gamma }$	0.000	0.000 ^a^	0.000	0.000
	$p_{\hat{\Phi },\Gamma }$	0.723	0.466	0.045	0.816

Superscript “a” Parameter is not normally distributed; the Wilcoxon Signed Ranks Test is used.

**Table 4 sensors-20-01903-t004:** The range of motion (ROM) with mean and standard deviation (SD) values calculated via the kinematic model Φ for Kinect v2, our deep learning refined kinematic model $\hat{\Phi }$ for Kinect v2 and the UWA model Γ for the 3DMC system. Φ−Γ represents the discrepancy between the ROMs via the model Φ and the reference model Γ. $\hat{\Phi }-\Gamma$ represents the differences between the ROMs via the model $\hat{\Phi }$ and the reference model Γ.

		Task 1	Task 2	Task 3	Task 4
**Shoulder Flexion/Extension (Degrees)**	Φ	51.38 (11.66)	87.53 (19.63)	146.30 (14.55)	38.04 (5.78)
	$\hat{\Phi }$	42.63 (7.73)	60.57 (15.25)	108.80 (13.44)	33.87 (4.71)
	Γ	42.50 (7.90)	58.83 (10.46)	105.85 (16.30)	34.15 (5.63)
	Φ−Γ	8.88 (4.89)	28.69 (12.50)	40.44 (9.88)	3.89 (2.76)
	$\hat{\Phi }-\Gamma$	0.13 (3.95)	1.74 (12.91)	2.95 (10.88)	−0.28 (3.19)
	$p_{\Phi ,\Gamma }$	0.000	0.000	0.000	0.000
	$p_{\hat{\Phi },\Gamma }$	0.905	0.636	0.348	0.422 ^a^
**Shoulder Adduction/Abduction (Degrees)**	Φ	26.26 (6.96)	18.76 (8.61)	40.37 (5.62)	18.02 (7.76)
	$\hat{\Phi }$	13.87 (3.88)	16.28 (8.48)	48.76 (5.53)	12.93 (5.84)
	Γ	14.08 (4.76)	13.71 (11.13)	47.36 (7.61)	12.97 (5.56)
	Φ−Γ	12.18 (5.33)	5.05 (5.32)	−6.98 (7.27)	5.05 (2.72)
	$\hat{\Phi }-\Gamma$	0.21 (2.97)	2.57 (5.00)	1.40 (4.86)	−0.04 (2.57)
	$p_{\Phi ,\Gamma \;}$	0.000	0.009 ^a^	0.000	0.000
	$p_{\hat{\Phi },\Gamma }$	0.802	0.075 ^a^	0.319 ^a^	0.861 ^a^
**Shoulder Internal/External Rotation (Degrees)**	Φ	35.16 (11.69)	19.34 (8.35)	56.44 (19.60)	15.57 (6.92)
	$\hat{\Phi }$	68.34 (6.47)	29.17 (6.24)	47.31 (15.41)	24.06 (4.26)
	Γ	64.59 (11.25)	30.07 (7.93)	45.69 (15.98)	22.92 (8.05)
	Φ−Γ	−29.42 (8.16)	−10.73 (8.47)	10.75 (10.39)	−7.35 (10.31)
	$\hat{\Phi }-\Gamma$	3.75 (9.33)	−0.90 (7.59)	1.62 (7.94)	1.14 (10.10)
	$p_{\Phi ,\Gamma \;}$	0.000	0.001	0.000	0.025
	$p_{\hat{\Phi },\Gamma }$	0.173	0.678	0.476	0.691
**Elbow Flexion/Extension (Degrees)**	Φ	87.54 (8.21)	88.70 (7.41)	87.57 (7.89)	41.28 (21.06)
	$\hat{\Phi }$	95.03 (5.77)	100.19 (6.39)	112.64 (4.22)	48.93 (12.68)
	Γ	95.03 (5.77)	100.29 (6.54)	112.97 (8.97)	47.82 (15.71)
	Φ−Γ	−7.49 (7.94)	−11.59 (6.57)	−25.41 (6.84)	−6.54 (7.74)
	$\hat{\Phi }-\Gamma$	0.92 (8.33)	−0.10 (6.52)	−0.34 (7.55)	1.11 (7.95)
	$p_{\Phi ,\Gamma \;}$	0.005	0.001	0.000	0.010
	$p_{\hat{\Phi },\Gamma }$	0.697	0.861	0.875	0.625

Superscript “a” Parameter is not normally distributed; the Wilcoxon Signed Ranks Test is used.

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
