# Peer review of "Deep Learning-Based Upper Limb Functional Assessment Using a Single Kinect v2 Sensor"

_sensors, 2020, doi:10.3390/s20071903_

Round 1

Reviewer 1 Report

This study presents a deep learning technique to improve the accuracy of Kinect data, a.k.a. markerless motion capture. This is a growing field in need of studies like this so the content is quite interesting. I don't have any recommendations to improve the technical content as the Method is sound. I only have some small recommendations to improve the organization and presentation:

--Small grammatical errors throughout the manuscript, including verb tenses.

--Some paragraphs are underdeveloped. Please add more content or re-organize so each paragraph has at least 4 sentences.

--The Introduction does a nice job of describing the current state of markerless motion capture and its need in the current rehabilitation market.

--The Methods are well done and easy to follow. The Figures are really helpful. However, some of the text of the figures is hard to read.

--The Results are presented well. Similar to the Methods, some of the text is difficult to read. Please enlarge.

--The Discussion does a nice job of explaining the Results within clinical applications.

Reviewer 2 Report

The paper proposes a framework to train a long short-term memory recurrent neural network using a supervised machine learning architecture to compensate for the systematic error of the Kinect kinematic model. However, revision should be needed to improve its current status before consideration for publishing :

1, The organization of the paper is clear, while its English wordings need a thorough revision;

2, Please explain the reason why to choose the long short-term memory (LSTM) recurrent neural network (RNN);

3, Page 7, Please explain the determined process for the cut-off frequency of the Butterworth low-pass filter;

4, The definition of UWA kinematic is repetitive.

Reviewer 3 Report

This manuscript describes a method to correct the systematic error of Kinect v2 sensor using deep learning. The overall methodology looks reasonable. However, the writing should be improved to explain the major contributions of this work in a better way. In addition, there are several issues that need to be addressed in experiment design.

  1. The researchers did not mention what measurement error is tolerable for different applications, so it is hard to tell if the proposed method works for these proposed applications.
  2. The inset of figure 1 is super small. The symbols mentioned in the beginning of section 2 should be indicated in figure to allow readers to understand the workflow better.
  3. There is a lot of redundancy in section 2.1 and 2.2. The methods mentioned in these two sections seem to be standard and straightforward. I would suggest authors to rewrite them and make them shorter to make the paper more coherent.
  4. What parameters are adjusted in the refined kinematic model compared with the kinematic model? Also, what is the systematic error and what factors cause the systematic error?
  5. Right before eq (14), should it be “forearm” instead of “upper arm”?
  6. In section 3, how many times are repeated for each task and each test subject?
  7. The authors mentioned that the measurement accuracy is plane-dependent. However, in section 3, only the frontal plane is evaluated. The authors should compare the results for different planes.
  8. The authors mentioned that the measurement accuracy is task-dependent. The authors should explain why these four tasks in Figure 6 are selected for training and testing. There are much more postures than these four.
  9. In section 4, the authors analyze the experimental results. However, the authors fail to explain why the proposed method works better for some postures than others.

Round 2

Reviewer 2 Report

I think it is good enough to publish now.

Author Response

We had another careful look at the whole manuscript and corrected the grammar mistakes and typos.

Reviewer 3 Report

The authors have addressed most issues found by reviewer 3. Only one comment as follows:

Question 6 is not answered. I do not see any revision regarding Q6 in section 3. Please say clearly how many times are repeated for each task and each human subject.

Author Response

We have carefully checked the phrases “each task” and “each human subject” in Section 3. We used “each task” three times. We revised this section and now there is no “each task” in this section. We highlighted this issue in the manuscript (line 358-359).

But after we carefully looked for “this subject”, we found none. Could you please point out where exactly is the “each human subject” (line number)?